# The Influence of *Aspergillus fumigatus* Fatty Acid Oxygenases PpoA and PpoC on Caspofungin Susceptibility

**DOI:** 10.3390/jof10110749

**Published:** 2024-10-30

**Authors:** Endrews Delbaje, Patrícia Alves de Castro, Dante G. Calise, Niu Mengyao, Maria Augusta Crivelente Horta, Daniel Yuri Akiyama, João Guilherme Pontes, Taícia Fill, Olaf Kniemeyer, Thomas Krüger, Axel A. Brakhage, Koon Ho Wong, Nancy P. Keller, Gustavo H. Goldman

**Affiliations:** 1School of Pharmaceutical Sciences of Ribeirão Preto, Universidade de São Paulo, Ribeirão Preto 14040-903, Brazil; endrews.delbaje@gmail.com (E.D.); patriciaalvesdecastro@yahoo.com.br (P.A.d.C.);; 2Department of Medical Microbiology and Immunology, University of Wisconsin-Madison, Madison, WI 53706, USA; dcalise@wisc.edu (D.G.C.); mengyao24@hotmail.com (N.M.); nancypulane@gmail.com (N.P.K.); 3Institute of Chemistry, State University of Campinas (Unicamp), Campinas 13083-862, Brazil; d195888@dac.unicamp.br (D.Y.A.); jgpontes@unicamp.br (J.G.P.); taicia@gmail.com (T.F.); 4National Institutes of Science and Technology in Human Pathogenic Fungi, Ribeirão Preto 14040-903, Brazil; 5Department of Molecular and Applied Microbiology, Leibniz Institute for Natural Product Research and Infection Biology (Leibniz-HKI), 07745 Jena, Germany; olaf.kniemeyer@leibniz-hki.de (O.K.); thomas.krueger@leibniz-hki.de (T.K.); axel.brakhage@hki-jena.de (A.A.B.); 6Department of Microbiology and Molecular Biology, Institute of Microbiology, Friedrich Schiller University, 07743 Jena, Germany; 7Faculty of Health Sciences, University of Macau, Taipa, Macau SAR 999078, China; koonhowong@um.edu.mo; 8Institute of Translational Medicine, Faculty of Health Sciences, University of Macaugrid, Avenida da Universidade, Taipa, Macau SAR 999078, China; 9MoE Frontiers Science Center for Precision Oncology, University of Macaugrid, Taipa, Macau SAR 999078, China; 10Department of Bacteriology, University of Wisconsin-Madison, Madison, WI 53706, USA

**Keywords:** *Aspergillus fumigatus*, fatty acid oxygenases, oxylipin, caspofungin

## Abstract

*Aspergillus fumigatus* can cause invasive pulmonary aspergillosis (IPA). Fungicidal azoles and fungistatic caspofungin (CAS) are the first- and second-line therapies, respectively, used to treat IPA. Treatment of *A. fumigatus* with CAS or micafungin induces the production of the oxylipin 5,8-diHODE by the fungal oxygenase PpoA. For this article, we investigated the influence of ppo genes, which encode the fatty acid oxygenases responsible for oxylipin biosynthesis, on CAS tolerance. The influence of PpoA and PpoC on CAS tolerance is mediated by MpkA phosphorylation and protein kinase A (PKA) activity. RNAseq transcriptional profiling and the label-free quantitative proteomics of the ppoA and ppoC mutants showed that differentially expressed genes and proteins are related to secondary metabolites and carbohydrate metabolism. We also characterized two clinical isolates, CM7555 and IFM61407, which decrease and increase susceptibility to CAS, respectively. CM7555 does not exhibit increased oxylipin production in the presence of CAS but oxylipin induction upon CAS exposure is increased in IFM61407, suggesting that oxylipins are not the only mechanism involved in CAS tolerance in these isolates. Upon CAS exposure, CM7555 has higher MpkA phosphorylation and PKA activity than IFM61407. Our results reveal the different aspects and genetic determinants involved in *A. fumigatus* CAS tolerance.

## 1. Introduction

*Aspergillus* species are saprophytic fungi that can be found ubiquitously in the environment. Within the large and diverse genus *Aspergillus*, a small number of species are known to cause infection in humans, with *A. fumigatus* being the most dangerous species that is responsible for more than 90% of *Aspergillus* infections. Like many other fungi, *A. fumigatus* can asexually produce great numbers of clonal spores (e.g., millions from a single colony) that are readily dispersed in the environment. It is estimated that we breathe in hundreds of spores from the environment each day. When inhaled, the spores of *A. fumigatus* can germinate and grow in the respiratory mucosa, leading to the development of invasive pulmonary aspergillosis (IPA). Aspergillosis can affect various organs by dissemination and disease can occur. IPA is the most severe form of *Aspergillus*-related infections, predominantly affecting individuals with hematological malignancies, genetic or acquired immunodeficiency, and patients receiving immunosuppressive therapy.

IPA incidence has risen substantially in the last few decades, due to a continuous increase in the number of high-risk populations, such as patients with malignant neoplasms, those undergoing organ transplantation, and patients receiving immunosuppressive treatments for various underlying conditions. The morbidity and mortality rates of IPA among these patients are extremely high, reaching over 90% [1]. During the COVID-19 pandemic, many COVID-19 patients were infected by *A. fumigatus* worldwide, with a mortality rate as high as 50% [2,3]. Therefore, individuals are becoming susceptible to this deadly *Aspergillus* infection after contracting COVID-19, which is now widely prevalent.

What makes the threatening situation worse is the fact that the treatment of *Aspergillus* infection (and fungal infection in general) is notoriously difficult. Currently, only three classes of drugs are available in the clinics to treat IPA (and fungal pathogens in general) without adverse side effects for the patients, contrasting with the two dozen antibacterial drugs available [4]. The main classes of antifungal drugs against IPA are azoles, polyenes, and echinocandins. Azoles such as fluconazole, itraconazole, posaconazole, and voriconazole are the first standard treatment against systemic fungal infections and are also the major drugs for IPA treatment, due to their broad spectrum of antifungal activity and good tolerance by humans [5]. Azole drugs act by inhibiting a key enzyme in the ergosterol biosynthetic pathway, Cyp51/Erg11, decreasing its content in the fungal cytoplasmic membrane and, thus, impairing fungal growth [6]. Polyene amphotericin B binds to different kinds of sterols, with higher affinity to ergosterol in the fungal cell membrane where it leads to pore formation, resulting in increased cell permeability and the leakage of small intracellular molecules [7,8]. In contrast, the echinocandin caspofungin (CAS), which is the second-line therapy against IPA when azole treatment is not effective, inhibits β-1,3-glucan synthase, thereby affecting β-1,3-glucan biosynthesis. Consequently, the composition and organization of the *A. fumigatus* cell wall is altered [9], thereby increasing osmotic sensitivity and causing growth inhibition (i.e., fungistatic effects). It is noteworthy that the clinical application of CAS has gained significant attention recently as a result of the rise of azole-resistant *A. fumigatus* isolates in clinics [10]. Paradoxically and interestingly, *A. fumigatus* can somehow survive and grow better at high CAS concentrations with normal morphology. This tolerance phenomenon, named the caspofungin paradoxical effect (CPE), also happens in vivo [11] and could greatly impair the effectiveness of CAS for patient treatment, rendering this second-line therapy (which is often the final resort for critical patients) ineffective. Moreover, similar paradoxical growth phenomena, albeit to a lesser extent, have also been observed with other echinocandins such as micafungin and anidulafungin [12,13].

The mechanism underlying CPE is not well understood. CPE is thought to be mediated by re-localized β-(1,3)-glucan synthase, reconstituted β-(1,3)-glucan synthesis, and normalized levels of cell wall chitin emerging from slow-growing microcolonies [14], although the precise mechanism behind CPE remains speculative. It is proposed that CPE depends on a complex network of interactions between the different pathways, which work together to allow *A. fumigatus* to resist and grow in the presence of high levels of CAS [15]. CPE can be triggered by an increase in intracellular Ca^2+^, which binds to calmodulin and activates the phosphatase calcineurin [16] to dephosphorylate the transcription factor CrzA. Dephosphorylated CrzA translocates to the nucleus and activates the genes for stress responses and cell wall modifications [17]. Ca^2+^ deprivation [18] and/or the absence of different components of the calcineurin pathway [19] result in the abolition of CPE in *A. fumigatus*. The mitochondrial respiratory chain also affects CPE, as ATP is required for the uptake of extracellular Ca^2+^ and activation of the Ca^2+^/calcineurin pathway [20]. In addition, the increased ATP production and oxygen consumption generates reactive oxygen species (ROS) in the mitochondria [21]. Mitochondrial ROS accumulation alters the plasma membrane lipid composition, causing a conformational change in the Fks1 enzyme that probably affects CAS binding, thereby restoring β-1,3-glucan synthase activity [22].

The cell wall integrity (CWI) pathway has also been implicated in controlling CAS tolerance and CPE. The CWI mitogen-activated protein kinase MpkA and its associated transcription factor RlmA regulate the expression of genes involved in chitin, β-1,3-glucan, and α-1,3-glucan biosynthesis in response to different concentrations of CAS [23]. Moreover, the SakA mitogen-activated protein kinase of the high-osmolarity glycerol (HOG) pathway is required for adaptation to CAS by activating cell-wall stress and via MpkA cross-talk [15]. Although these signaling pathways have been shown to affect CPE in *A. fumigatus*, their downstream targets, regulatory mechanisms, and the interactions between the pathways have not yet been fully elucidated. Recently, we showed that treatment of *A. fumigatus* with CAS or micafungin, but not with voriconazole, induces the production of the oxylipin 5,8-diHODE by the fungal oxygenase PpoA [24]; 5,8-diHODE is produced by the dihydroxylation of linoleic acid, first to 8(*R*)-hydroxy-9Z,12Z70 octadecadienoic acid (8-HODE) and, subsequently, to 5,8-diHODE by the linoleate diol synthase PpoA [25]. Another fatty acid oxygenase, PpoC, produces another linoleic acid-derived oxylipin, 10-hydroxyoctadecadienoic acid (10-HODE). Besides *ppoA* and *ppoC*, *ppoB* encodes another uncharacterized fatty acid oxygenase. For this study, we analyzed the involvement of *ppo* genes in CAS tolerance. We demonstrated that although oxylipins are important for CAS tolerance, other genetic determinants are also involved in this phenomenon. Our findings on CAS tolerance and CPE may shed light on how fungi cope with different kinds of stress and may have implications for combating *A. fumigatus* and other human fungal pathogens.

## 2. Materials and Methods

### 2.1. Strains and Media

The *A. fumigatus* strains were grown at 37 °C in minimal medium (MM: 1% (*w*/*v*) glucose, original high nitrate salts, trace elements, pH 6.5). Solid MM was the same as described above, except that 2% (*w*/*v*) agar was added. Trace elements and nitrate salt compositions were as described previously [26]. For phenotype characterization, plates were inoculated with 10^4^ spores per strain and left to grow for 120 h at 37 °C. All radial growths were expressed as ratios, dividing the colony radial diameter of growth in the stress condition by the colony radial diameter in control (no stress) conditions.

### 2.2. Mutant Construction

Three fungal mutants were newly constructed in this study, including the double deletion mutant Δ*ppoA* Δ*ppoB* (TMN30), and overexpression mutants OE::*ppoB* (TMN33) and OE::*ppoC* (TMN34). Δ*ppoA* Δ*ppoB* (TMN30) was created by deleting ppoA and ppoB sequentially on Af293.6 (*pyrG1*, *argB1*) [27]. The *ppoA* (Afu4g10770) deletion construct was amplified from pDWC4.2 [28] and then transformed into Af293.6 (*argB1*, *pyrG1*) to create the *∆ppoA argB* auxotroph. This strain was subsequently used to delete the *ppoB* gene (Afu4g00180) with the *A. fumigatus argB* marker amplified from pJMP4 [29], leading to TMN30. A deletion cassette for *ppoB* was constructed by fusing the ~1 kb 5′ and 3′ flanking regions of the gene with *A. parasiticus pyrG*. Overexpression strains of *ppoB* (TMN33) and *ppoC* (TMN34) were achieved by inserting the selectable marker *A. parasiticus pyrG* and *A. nidulans gpdA* promoter upstream of the *ppoB* or *ppoC* (Afu3g12120) gene through homologous recombination in the parental strain Af293.1(*pyrG1*) [30]. The ~1 kb sequence upstream and downstream of the translation start site was amplified as the 5′ and 3′ flanking regions, respectively. The pyrG::gpdA(p) sequence was amplified from pJMP9.1 [31]. All the overexpression fusions happened at the translational start site.

DNA transformation constructs were created through double-joint PCR using the previously published protocol [32]. Protoplast generation and transformation were performed according to the previously published protocol [33]. All transformants were first screened through PCR for incorporation of the construct and the absence of the *ppo* gene (deletion strain only). Southern blotting, followed by the hybridization of αP^32^-dCTP-labeled 5′ and 3′ flank regions, were used to confirm those transformants with single integration (Appendix A; https://doi.org/10.6084/m9.figshare.27134901).

The overexpression of *ppoB* and *ppoC* genes was confirmed through semi-quantitative reverse transcription-PCR (semi-q RT-PCR). WT, OE::*ppoB*, and OE::*ppoC* spores were grown in liquid GMM at 37 °C and 250 rpm for 24 hr. Fungal biomass was collected, flash-frozen in liquid nitrogen, lyophilized, and extracted for total RNA using QIAzol lysis reagent (Qiagen, Venlo, The Netherlands), according to the manufacturer’s instructions. RNA was assessed while in polyacrylamide gel for quality confirmation. Total RNA was treated with DNaseI (New England Biolabs, Ipswich, MA, USA) before cDNA synthesis, using an iScript kit (Bio-Rad, Hercules, CA, USA). For this study, 50 ng of cDNA was used for all PCR reactions, using in-house Taq polymerase extract and the Green GoTaq^®^ Reaction Buffer (Promega, Madison, WI, USA). The Af293 WT genomic DNA and ddH_2_O were used as the positive and negative controls, and actin gene expression was used as the reference control. PCR products were loaded in a 2% agarose gel, separated through gel electrophoresis, and imaged under UV light. Primer pairs are listed in Appendix A (https://doi.org/10.6084/m9.figshare.27134901) and the Semi-q RT-PCR results are also illustrated in the Appendix A, https://doi.org/10.6084/m9.figshare.27134901).

### 2.3. Western Blot Analysis

Wild-type *A. fumigatus*, Δ*ppoA*, and Δ*ppoC* were grown in liquid MM for 16 h at 37 °C, and were transferred or not to 2 µg/mL of CAS for 0, 10, 30, and 60 min. The OE*ppoA* and OE*ppoC* were grown in liquid MM for 16 h at 37 °C. Total cellular protein extractions were carried out as described previously [34], then quantified using Bradford reagent (Bio-Rad, Hercules, CA, USA) according to the manufacturer’s instructions. Fifty µg of protein from each sample was resolved in a 12% (*w*/*v*) SDS–PAGE and transferred to polyvinylidene difluoride (PVDF) membranes (Merck Millipore, Burlington, MA, USA). The phosphorylated fractions of the MAP kinase, MpkA, and total MpkA were identified using P-p44/42 and p44/42 antibodies (Cell Signaling Technologies, Danvers, MA, USA), respectively, following the manufacturer’s instructions and using a 1:2000 dilution). The primary antibody was detected using an HRP-conjugated secondary antibody, raised in rabbit (Sigma, St. Louis, MA, USA). Chemoluminescent detection was achieved using an ECL Prime Western blot detection kit (GE HealthCare, Little Chalfont, UK). To detect these signals on blotted membranes, the ECL Prime Western blotting detection system (GE Healthcare, Little Chalfont, UK) and LAS1000 (FUJIFILM, Tokyo, Japan) were used.

### 2.4. PKA Activity Assay

The *A. fumigatus* strains were grown in liquid MM for 16 h at 37 °C and then transferred or not to 2 µg/mL of CAS for 0, 10, 30, and 60 min. The total protein content was extracted by grinding the pellets using a mortar and pestle in liquid nitrogen and resuspending the results in the lysis buffer present in the kit, supplemented with an EDTA-free Protease Inhibitor Cocktail tablet (Roche), phenyl methane sulfonyl fluoride (PMSF, 1 mM final concentration) and 50 μL of activated sodium orthovanadate (10 mM final concentration). The extracts were centrifuged at 13,000× *g* for 20 min at 4 °C. The supernatants were collected and the total protein was quantified using Bradford reagent (Bio-Rad). The PKA (Protein Kinase A) Colorimetric Activity Kit (EIAPKA, ThermoFisher Scientific, Waltham, MA, USA) was utilized according to the manufacturer’s instructions. The endpoint reaction was detected using a Synergy-HT microplate reader (Bio-Tek) at 450 nm. PKA activity was determined relative to the total protein content of the samples.

### 2.5. Genome Assembly of CM7555 and IFM61407

The public genome data were obtained from the NCBI SRA database (strain CM7555 with accession DRX013578; strain IFM61407, with accession SRR7418938). The raw data quality and the presence of adapters were visualized with FastQC v0.11.8 [35]. The adapter sequences were trimmed, and reads were quality-filtered using Trimmomatic v0.39 [36] with the parameters ‘LEADING:20 TRAILING:20 MINLEN:60′. For the de novo genome assembly, the program SPAdes v.3.14.0 [37] was used. The assembly statistics (genome length, contigs, GC(%), and N50) were evaluated with QUAST v5.2.0 [38].

### 2.6. RNA Extraction, RNA-Sequencing and RTqPCR

Strains were grown for 16 h in liquid MM at 37 °C and then exposed or not to CAS 2 µg/mL for 1 h. OE*ppoA*, and OE*ppoC* were grown for 16 h in liquid minimal medium without any further exposure to CAS. The mycelia were harvested and frozen in liquid nitrogen. Total RNA was extracted using the Trizol method (Invitrogen, Waltham, MA, USA), treated with RNase-free DNase I (Fermentas, Waltham, MA, USA), and purified using an RNAeasy Kit (Qiagen, Venlo, Netherlands) according to the manufacturer’s instructions. The RNA from each exposure was quantified using a NanoDrop and Qubit fluorometer and analyzed using an Agilent 2100 bioanalyzer system to assess the integrity of the RNA. The RNA integrity number (RIN) was also calculated; the RNA sample had an RIN of  9.0–9.5. For library preparation, the Illumina TruSeq Stranded mRNA Sample Preparation kit was used according to the manufacturer’s protocol. For RTqPCR, the RNA was reverse-transcribed to cDNA using the ImProm-II reverse transcription system (Promega), and the synthesized cDNA was used for real-time analysis using the SYBR green PCR master mix kit (Applied Biosystems) in the ABI 7500 fast real-time PCR system (Applied Biosystems, Foster City, CA, USA). The primer sequences used for the *ppoA* and *ppoC* genes are described in Appendix A (https://doi.org/10.6084/m9.figshare.27134901).

### 2.7. Differential Expression Analysis

The quality of sequences was assessed using FastQC v0.11.8 [35]. The adapter sequences were trimmed, then reads were quality-filtered using Trimmomatic v0.39 [36], with the parameters ‘ILLUMINACLIP:NexteraPE-PE.fa:2:30:10 LEADING:20 TRAILING:20 MINLEN:60′. rRNA reads were removed using SortMeRNA v4.3.6 [39]. The filtered reads were then aligned to the *A. fumigatus* reference sequence Af293 (NCBI assembly code GCA_000002655.1) using Hisat v2.2.1 [40], with splice site positions and a maximum intron length of 3093, both of which were obtained from the reference Af293. Subsequently, the SAM files were converted to BAM format and sorted using SAMtools v1.15.1 [41]. FeatureCounts v1.5.3 [42] was used to quantify the non-ambiguous aligned fragments and to produce the tables for RNAseq analysis. For the differential expression analysis, the counts were converted to counts per million (CPM). The features were filtered, ensuring that the sum of all samples had a CPM of at least 3, considering those samples with a CPM above 1 for the sum. Differential expression analysis of the filtered genes was performed using edgeR v3.14.0 [43] with the simple design protocol [44].

### 2.8. Proteome Analysis

*Aspergillus fumigatus* Af293 and the *ppoA* and *ppoC* mutants were analyzed in triplicate, each without treatment (DMSO control) and with caspofungin treatment. The harvested cells were ground with a mortar and pestle in liquid nitrogen and lysed in 1% SDS, 150 mM NaCl, and 100 mM triethylammonium bicarbonate (TEAB), with the addition of 1 tablet of complete ultra protease inhibitor cocktail and PhosStop (both Roche) per 10 mL of lysis buffer. Subsequently, the cysteine thiols were reduced and carbamidomethylated in one step for 30 min at 70 °C by the addition of 2 µL of 500 mM TCEP (tris(2-carboxyethyl)phosphine) and 2 µL of 625 mM 2-chloroacetamide (CAA) per 100 µg of total protein in each 100 µL. The proteins were precipitated by MeOH/chloroform/H_2_O [45] and resolubilized in 100 mM TEAB in 5:95 trifluoroethanol/H_2_O (*v*/*v*). The proteins were digested for 18 h at 37 °C after the addition of Trypsin/Lys-C mix at a protease-to-protein ratio of 1:25. Tryptic peptides were dried in a vacuum concentrator (Eppendorf) and resolubilized in 30 µL of 0.05% TFA in H_2_O/ACN 98/2 (*v*/*v*), filtered through 10 kDa MWCO PES membrane spin filters (VWR). The filtrate was transferred to HPLC vials and injected into the LC-MS/MS instrument.

Each sample was measured in triplicate (3 analytical replicates of 3 biological replicates). LC-MS/MS analysis was performed on an Ultimate 3000 nano RSLC system connected to an Orbitrap Exploris 480 mass spectrometer (both Thermo Fisher Scientific, Waltham, MA, USA) with FAIMS. Peptide trapping was conducted for 5 min on an Acclaim Pep Map 100 column (2 cm × 75 µm, 3 µm) at 5 µL/min, followed by separation on an analytical Acclaim Pep Map RSLC nano column (50 cm × 75 µm, 2 µm). The mobile phase gradient elution of eluent A (0.1% (*v*/*v*) formic acid in water), mixed with eluent B (0.1% (*v*/*v*) formic acid in 90/10 acetonitrile/water), was performed using the following gradients: 0 min at 4% B, 20 min at 6% B, 45 min at 10% B, 75 min at 16% B, 105 min at 25% B, 135 min at 45% B, 150 min at 65% B, 160–165 min at 96% B, 165.1–180 min at 4% B. Positively charged ions were generated at a spray voltage of 2.2 kV using a stainless steel emitter attached to the Nanospray Flex ion source (Thermo Fisher Scientific). The quadrupole/orbitrap instrument was operated in Full MS/data-dependent MS2 mode. Precursor ions were monitored at *m*/*z* 300–1200 at a resolution of 120,000 FWHM (full width at half maximum), using a maximum injection time (ITmax) of 50 ms and a 300% normalized AGC (automatic gain control) target. Precursor ions with a charge state of z = 2–5 were filtered at an isolation width of *m*/*z* 4.0 amu for further fragmentation at 28% HCD collision energy. MS2 ions were scanned at 15,000 FWHM (ITmax = 40 ms, AGC= 200%). Three different compensation voltages were applied (−48 V, −63 V, and −78 V).

### 2.9. Protein Database Search

Tandem mass spectra were searched against the UniProt pan proteome databases (2023/01/16) of *Aspergillus fumigatus* (https://ftp.uniprot.org/pub/databases/uniprot/current_release/knowledgebase/pan_proteomes/UP000002530.fasta.gz; accessed on 21 December 2022) using Proteome Discoverer (PD) 3.0 (Thermo) and the database search algorithms Mascot 2.8, Comet, MS Amanda 2.0, Sequest HT, with and without INFERYS rescoring, and CHIMERYS. Two missed cleavages were allowed for the tryptic digestion results. The precursor mass tolerance was set to 10 ppm and the fragment mass tolerance was set to 0.02 Da. Modifications were defined as dynamic Met oxidation, the phosphorylation of Ser, Thr, and Tyr, and protein N-term acetylation with and without Met-loss, as well as static Cys carbamidomethylation. A strict false discovery rate (FDR) of <1% (peptide and protein levels) was required for positive protein hits. Furthermore, search engine scores were filtered for either Mascot (>30), Comet (>3), MS Amanda 2.0 (>300), Sequest HT (>3), or CHIMERYS (>2). The Percolator node of PD3.0 and a reverse decoy database were used for the q-value validation of spectral matches. Only rank-1 proteins and the peptides of the top-scoring proteins were counted. Label-free protein quantification was based on the Minora algorithm of PD3.0, using precursor abundance based on intensity and a signal-to-noise ratio of >5. Normalization was performed using the total peptide amount method. The imputation of missing quantification values was applied, using abundance values of 75% of the lowest abundance identified. Differential protein abundance was defined as a fold change of >2, a *p*-value/ABS (log4ratio) of <0.05, and at least being identified in 2 of the 3 replicates of the sample group with the highest abundance.

### 2.10. HPLC-HRMS^2^ Analysis

HPLC-HRMS^2^ positive-mode analysis was performed in a Thermo Scientific QExactive Hybrid Quadrupole-Orbitrap mass spectrometer, coupled to a Dionex UltiMate 3000 RSLCnano system. As the stationary phase, a Thermo Scientific column Accucore C18 2.6 µm (2.1 mm × 100 mm × 1.7 µm) was used. The mobile phase was 0.1% formic acid (A) and acetonitrile (B). Eluent profile (A/B %): 95/5 for 5 min, reaching 60/40 at 10 min, up to 55/45 within 2 min and reaching 2/98 at 18 min, maintained for 2 min, down to 95/5 within 2 min, and maintained for 2 min. Total run time was 24 min for each run at a flow rate of 0.3 mL.min^−1^. The injection volume was 5 µL. MS spectra were acquired, with *m*/*z* ranges from 100 to 1500, and 70,000 mass resolution at 200 Da. Ionization parameters: sheath gas flow rate (45), aux gas flow rate (10), sweep gas flow rate (2), spray voltage (3.5 kV), capillary temperature (250 °C), S-lens RF level (50), and auxiliary gas heater temperature (400 °C). MS^2^ spectra were acquired using the data-dependent acquisition (DDA) mode. The normalized collision energy was applied stepwise (20, 30, and 40 eV), and the 5 most intense precursors per cycle were measured with 17,500 mass resolution at 200 Da.

### 2.11. Secondary Metabolite Annotation

Raw HPLC-HRMS^2^ data were converted into mzML format files in MSConvert, with 32-bit binary encoding precision, zlib compression, and peak peaking. Secondary metabolites were annotated in the online workflow (https://ccms-ucsd.github.io/GNPSDocumentation/, accessed on 15 March 2023) on the GNPS website (http://gnps.ucsd.edu, accessed on 15 March 2023). The data were filtered by removing all MS/MS fragment ions within +/- 17 Da of the precursor *m*/*z*. MS/MS spectra were window-filtered by choosing only the top 6 fragment ions in the +/- 50 Da window throughout the spectrum. The precursor ion mass tolerance was set to 0.02 Da at an MS/MS fragment ion tolerance of 0.02 Da. The spectra were then searched against the GNPS spectral libraries. The library spectra were filtered in the same manner as the input data. All matches that were kept between the network spectra and library spectra were required to have a score above 0.7 and at least 5 matched peaks.

### 2.12. Fungal Oxylipin Extraction

The *A. fumigatus* conidia were inoculated at 5 × 10^6^ spores per mL into 50 mL GMM and incubated at 37 °C and 250 RPM for 48 h before the addition of caspofungin (2 µg/mL) or DMSO (0.0002%, *v*/*v*) and an additional 48-hour incubation. The supernatants were filtered into 250 mL glass bottles by collecting mycelial tissue in sterile miracloth. The collected tissue was press-dried, flash-frozen, and lyophilized. Supernatants were extracted overnight for oxylipins, as previously described, with 100 mL ethyl acetate (EA):methanol (MeOH):dichloromethane (DCM) (8:1:1) mixed organic solvent. The organic phase was collected using a separatory funnel and evaporated to dryness using a Buchi Rotovap R-210. The lyophilized tissue was weighed to determine the dry biomass before being homogenized in 10 mL sterile water with 0.01% formic acid. Homogenized tissue was extracted overnight in 120 mL mixed organic solvent at EA:MeOH:DCM. The tissue and debris were filtered out using Whatman paper filters, and an equal volume of Milli-Q water was added to the mixed organic solvent in a separatory funnel. The organic layer was collected after shaking and venting twice and evaporated to dryness. The total extracts were dissolved in 4 mL MeOH and transferred to pre-weighed 20 mL glass scintillation vials, then evaporated to dryness again. The dried extracts were weighed and stored at −80 °C. Extracts were dissolved at 1 mg/mL in MeOH for analysis by UHPLC–HRMS/MS.

### 2.13. UHPLC–HRMS/MS Analysis

The ultra-high-pressure liquid chromatography–high-resolution mass spectrometry (UHPLC–HRMS) data were acquired using a Thermo Scientific Q Exactive Orbitrap mass spectrometer coupled to a Vanquish UHPLC and operated in both positive and negative ionization modes. All the solvents used were of spectroscopic grade. Each sample was filtered with a 0.2 µm syringe filter. A Waters XBridge BEH-C18 column (2.1 mm × 100 mm, 1.7 μm) was used with acetonitrile (0.1% formic acid) and water (0.1% formic acid) as solvents, at a flow rate of 0.2 mL/min. The screening gradient method for the samples was as follows: starting at a 55% organic hold for 1 min, followed by a linear increase to a 98% organic hold for 18 min, then holding at a 98% organic hold for 2 min, comprising a total of 21 min. A quantity of 10 μL of each sample was injected into the experimental system for the analysis. Purified 5,8-diHODE, 8-HODE, and 10-HODE were used as standards. For the quantification, standard curves for 5,8-diHODE, 8-HODE, and 10-HODE were calculated based on the intensities from 6 different concentrations of each purified oxylipin (5, 2.5, 1.25, 0.625, 0.3125, and 0.15625 ppm).

## 3. Results

### 3.1. A. fumigatus ppo Mutants Have Altered Caspofungin Susceptibility

There are three *ppo* genes that encode fatty acid oxygenases in *A. fumigatus* [46]. We characterized the CAS susceptibility of Δ*ppoA*, Δ*ppoB*, Δ*ppoC*, the double mutants Δ*ppoA* Δ*ppoB*, Δ*ppoA* Δ*ppoC*, and Δ*ppoB* Δ*ppoC*, and the constitutively overexpressed *gpdA::ppoA* (OE*ppoA*), *gpdA::ppoB* (OE*ppoB*), and *gpdA::ppoC* (OE*ppoC*) strains (Figure 1a,b). At low CAS concentrations (0.250 µg/mL), Δ*ppoA*, Δ*ppoB*, Δ*ppoC*, the double mutants Δ*ppoA* Δ*ppoB*, Δ*ppoA* Δ*ppoC*, and OE*ppoB* showed higher CAS sensitivity than the wild-type strain (Figure 1a). In contrast, OE*ppoA* and OE*ppoC* were more resistant to CAS 0.250 µg/mL than the wild-type strain (Figure 1a). At high concentrations, CSP induced a tolerance phenotype with the partial reestablishment of fungal growth, called the CAS paradoxical effect (CPE), resulting from a change in the composition of the cell wall (for a review, see Ref. [20]). Surprisingly, the double mutant Δ*ppoB* Δ*ppoC* is as sensitive to CAS as the wild type, while Δ*ppoB* and Δ*ppoC* are more sensitive than the double mutant (Figure 1a). It is possible that there are additional mechanisms for coping with CAS stress that are activated when both genes are deleted; this remains to be determined.

Surprisingly, all the mutants had reduced CPE (Figure 1b). We also assessed the susceptibility of these mutants to cell wall damaging agents, such as Congo Red (CR) and Calcofluor White (CFW), and the highly osmotic conditions provided by 1.2 M sorbitol (Figure 1c,d). The Δ*ppoB*, all the double mutants, and OE*ppoB* were more resistant to cell wall damaging agents and osmotic stress conditions (Figure 1c,d).

As an initial step to understand the mechanisms of altered caspofungin susceptibilities that are dependent on *ppo* mutants, we decided to concentrate our attention on the four mutants of Δ*ppoA*, Δ*ppoC*, OE*ppoA*, and OE*ppoC* in subsequent experiments. It has previously been shown that CAS induces *A. fumigatus* MAP kinase MpkA from the cell wall integrity pathway and protein kinase A activity (PKA) [47,48,49]. Exposure of wild-type *A. fumigatus* to 2 µg/mL CAS for different periods of time showed a peak of MpkA phosphorylation at 10 min, while the overexpression of PpoA and PpoC for 16 h showed two-fold higher MpkA phosphorylation than the wild type (Figure 2a).

In contrast, the exposure of Δ*ppoA* and Δ*ppoC* to CAS showed the delayed induction of and no MpkA phosphorylation, respectively (Figure 2a). Upon exposure to CAS (2 µg/mL) for different periods of time, increased activity from PKA (up to 3-fold) was seen in the wild-type strain (Figure 2b). Increased PKA activity was also observed when PpoA and PpoC were overexpressed for 16 h (Figure 2b). However, there was a significant decrease in PKA activity in the Δ*ppoA* and Δ*ppoC* mutants when compared to the wild type (Figure 2b).

These results suggest that the *ppo* mutants are important for the response to CAS. Moreover, PpoA and PpoC influence CAS susceptibility, which is probably mediated by MpkA phosphorylation and PKA activity.

### 3.2. Transcriptional Profiling of PpoA and PpoC Null and Overexpression Mutants

To identify possible gene targets that are affected by *ppoA*/*ppoC* deletion and/or overexpression, we performed RNAseq transcriptional profiling. In the experimental design, the wild-type, Δ*ppoA*, and Δ*ppoC* strains were grown in the liquid minimal medium for 16 h at 37 °C and then exposed to CAS 2 µg/mL for 1 h. Alternatively, the wild-type, OE*ppoA*, and OE*ppoC* were grown for 16 hrs in liquid minimal medium without any further exposure to CAS. The OE*ppoA* and OE*ppoC* have about 12− and 8-fold more *ppoA* mRNA accumulation and 17- and 6-fold more (at 16 and 24 h), respectively, than the wild type (Appendix A). I*n* both experimental designs, we focused on the differentially expressed genes (DEGs) that have log2 fold changes of ≥1 and ≤−1 (an FDR of 0.05). Surprisingly, we were not able to identify any DEGs when comparing Δ*ppoA* and OE*ppoA* with the wild-type strain grown under the same conditions. In contrast, Δ*ppoC* showed 52 and 68 DEGs that were down- and upregulated, respectively, in the presence of CAS. FunCat (Functional Catalog) (https://elbe.hki-jena.de/fungifun/fungifun.php, accessed on 29 September 2024) analyses for these DEGs [50] showed the transcriptional upregulation of genes coding for the proteins involved in secondary metabolism and non-ribosomal peptide synthesis, C-compound and carbohydrate transport, and cellular import (*p*-value < 0.01; Figure 2c and Appendix A, https://doi.org/10.6084/m9.figshare.27134901). FunCat analysis for the downregulated genes showed enrichment for those genes also involved in secondary metabolism (*p*-value < 0.01; Figure 2c). When comparing the wild type with the *OEppoC* grown for 16 h without exposure to CAS, a total of 312 genes were downregulated and 580 genes were upregulated. FunCat analysis for these DEGs showed the transcriptional up- and downregulation of the genes coding for proteins involved in secondary metabolism and C-compound and carbohydrate transport (*p*-value < 0.01; Figure 2d and Appendix A, https://doi.org/10.6084/m9.figshare.27134901). We did not observe *ppoA*, *ppoB*, and *ppoC* differential expression in either Δ*ppoC* or OE*ppoC* DEGs (Appendix A), suggesting that PpoC-altered levels do not affect the differential expression of these two genes.

We evaluated the impact of *ppoC* overexpression on secondary metabolite production by analyzing 5-day-cultured supernatants (grown at 37 °C) of the wild-type and OE*ppoC* strains by liquid chromatography–high-resolution mass spectrometry (LC-HRMS) analysis (Figure 2e). Several SMs, such as fumagillin, pseurotin A, fumigaclavine C, bisdethiobis(methylthio)gliotoxin, verruculogen, brevianamide F, and fumitremorgin C, showed increased production by the OE*ppoC* strain when compared to the wild-type strain; surprisingly, no SM production had decreased in the OE*ppoC* strain (Figure 2e).

Taken together, our transcriptomics and metabolomics analyses suggest that PpoC has a great impact on secondary metabolism. Previous studies in other *Aspergillus* spp., including *A. nidulans* and *A. flavus* [51,52], support a role for Ppo proteins in secondary metabolism.

### 3.3. PpoA and PpoC Share Common and Unique Proteomic Signatures

Our transcriptomic results with Δ*ppoA* and OE*ppoA* suggest that the main mode of action of oxylipins produced by PpoA is not at the transcriptional level. Accordingly, we performed proteomic analysis by growing the wild-type, Δ*ppoA*, and Δ*ppoC* strains in liquid MM + 0.1% (*w*/*v*) yeast extract for 16 h at 37 °C and adding either 2 µg/mL of caspofungin dissolved in DMSO or only DMSO (control) for an additional 2 h. We used label-free quantitative proteomics to investigate proteins that are differentially abundant in the Δ*ppoA* and Δ*ppoC* mutants after caspofungin stress. Differentially expressed proteins were defined as those with a minimum two-fold change in protein abundance (log_2_FC ≥ 1.0 and ≤−1.0; FDR of 0.05) when compared to the wild-type strain in the presence of CAS. There were 477 and 86 proteins that were up- and downregulated in Δ*ppoA*, while 571 and 145 proteins were up- and downregulated in Δ*ppoC*, respectively (Figure 3a). Venn diagram plotting of the proteins differentially expressed in Δ*ppoA* and Δ*ppoC* versus the wild-type strain in the presence of CAS revealed that the mutants shared 375 and 65 up- or downregulated proteins, respectively (Figure 3a). Mutant-specific proteomics responses showed 101 and 21 unique up- or downregulated proteins for Δ*ppoA* and 196 and 80 unique up- or downregulated proteins for Δ*ppoC* (Figure 3a). FunCat enrichment analysis for downregulated proteins in both mutant strains did not provide any enrichment; however, upregulated proteins in the Δ*ppoA* and Δ*ppoC* mutants showed enrichment for secondary metabolism and C-compound and carbohydrate metabolism (Figure 3b).

We went through the list of proteins that were modulated either in both mutants or specifically in each mutant (Table 1, Figure 3c). The analysis of proteins concomitantly up-or downregulated in both the Δ*ppoA* and Δ*ppoC* mutants upon CAS stress revealed proteins involved in (i) calcium metabolism, such as AFUA_2G07630 (encoding a vacuolar calcium ion transporter) and AFUA_2G05325 (encoding a calcium/proton exchanger); (ii) cell wall metabolism, such as with AFUA_5G02290 (encoding an endo-chitinase), AFUA_4G09900 (encoding a cell wall biogenesis protein Ecm15), and AFUA_1G17250 (encoding a hydrophobin); (iii) carbohydrate metabolism, such as AFUA_4G14090 (encoding a UDP-glucose-4-epimerase) and AFUA_2G01840 (encoding a phosphoglycerate mutase); (iv) the regulation of processes (for example, transcription factors and signal transduction proteins), such as with AFUA_3G08010 (encoding a C_2_H_2_ transcription factor Ace1) and AFUA_4G10110 (encoding HtfA, a homeobox domain-containing protein); and (v) iron metabolism, such as with AFUA_3G03390, AFUA_3G03400, and AFUA_1G17190 (encoding SidJ, SidF, and SidI, respectively, which are important for siderophore biosynthesis) (Figure 3c).

The analysis showed that protein up- and downregulation, specifically in Δ*ppoA* and Δ*ppoC*, was revealed in Δ*ppoA*, AFUA_7G01430 (encoding NopA, opsin), AFUA_1G02800 (encoding a chitinase), AFUA_2G05730 (encoding an MFS siderochrome iron transporter C), and in Δ*ppoC*, AFUA_8G00230 (encoding a verruculogen synthase), AFUA_7G04800 (encoding a G-protein coupled receptor GprC), and AFUA_4G06940 (encoding a sphingolipid-4-desaturase) (Table 1).

We tested selected mutants for transcription factors (TFs), a protein phosphatase, and two GPCRs for their susceptibility to inhibitory and CPE CAS concentrations of 0.25 µg/mL and 8.0 µg/mL, respectively (Figure 4a,b). We observed that the TF mutants AFUA_2G02540, AFUA_5G00290, AFUA_5G07510, AFUA_4G10340, AFUA_6G07010, AFUA_1G11000, and AFUA_1G16410 were more resistant to both CAS concentrations of 0.25 µg/mL and 8.0 µg/mL (Figure 4a,b). In contrast, AFUA_2G05310 and AFUA_4G10110 were more resistant to only 0.25 µg/mL of CAS, and AFUA_4G09710 was only more resistant to CAS 8.0 µg/mL (Figure 4a,b). We have identified that Δ*ppsA* (*ppsA* encodes a protein phosphatase that is the homolog of *Saccharomyces cerevisiae PPS1* protein phosphatase, which plays a role in the DNA synthesis phase of the cell cycle (https://www.yeastgenome.org/locus/S000000480, accessed on 29 September 2024), being more susceptible to both CAS concentrations of 0.25 µg/mL and 8.0 µg/mL (Figure 4a,b).

Taken together, these results suggest that upon CAS stress, PpoA and PpoC affect the accumulation of the proteins involved in the cell wall, carbohydrate, calcium, and SM metabolism. Several TF mutants and a protein phosphatase were validated as being involved in CAS susceptibility.

### 3.4. Molecular Characterization of Two Clinical Isolates That Have Contrasting Responses to Caspofungin: CM7555 and IFM61407

We have previously reported the occurrence of two clinical *A. fumigatus* isolates, CM7555 and IFM61407, which exhibit decreased and increased susceptibility to CAS, respectively [53,54] (Figure 5a). CM7555 is more resistant to CR and CFW than IFM61407, but both showed the same susceptibility to osmotic stress induced by increasing concentrations of sorbitol (Figure 5b,c). Both clinical isolates showed comparable *ppoA* and *ppoC* mRNA accumulation when they were exposed to different CAS concentrations (Figure 5d,e). We previously reported that the clinical isolate Af293 showed increased accumulation of 5-diHODE and 8-HODE when exposed to CAS [24] (Figure 5f,g). We evaluated whether the differential CAS susceptibility of CM7555 and IFM61407 could be linked to the production of a different oxylipin in the presence of CAS (Figure 5g,h). IFM61407 had increased levels of 5,8-diHODE and 8-HODE when exposed to CAS 2.0 µg/mL (Figure 5h,i). CM7555 showed slightly increased production of 5,8-diHODE but comparable 8-HODE when exposed to CAS (Figure 5h,i). A longer time of exposure to CAS induced much higher MpkA phosphorylation in CM7555 than in IFM61407 (Figure 5j), and the same behavior was also observed for PKA activity (Figure 5k).

To identify the possible differences between CM7555 and IFM61407, we assembled the genomes of these two strains and also performed RNAseq transcriptional profiling. Both strains had genome sizes, GC content, and mitochondrial genomes that were comparable to the Af293 reference strain (Table 2).

Single-nucleotide polymorphism (SNP) analysis revealed 20 SNPs in 17 genes when CM7555 was compared to IFM61407; Af293 is used as a reference genome strain (Table 3). None of these genes had previously been identified as involved in fungal drug resistance or, more specifically, in CAS tolerance/resistance. Therefore, additional studies had to be performed to validate their involvement in these processes.

For the RNAseq experimental design, CM7555 and IFM61407 clinical isolates were grown in liquid minimal medium for 16 h at 37 °C and then exposed to CAS 2 µg/mL for 1 h. We concentrated our attention on differentially expressed genes (DEGs) that have a log2 fold of ≥1 and ≤1 (FDR of 0.05) when we compared CM7555 with the IFM61407 strains. CM7555, compared to the IFM61407 strains, showed 419 and 216 DEGs down- and upregulated, respectively, in the absence of CAS, and 936 and 370 DEGs down- and upregulated, respectively, in the presence of CAS. FunCat analysis for these DEGs in the absence of CAS showed the transcriptional downregulation of genes coding for proteins involved in C-compound and carbohydrate metabolism and secondary metabolism and in the C-compound and carbohydrate transport and the upregulation of secondary metabolism and siderophore-iron transport (*p*-value < 0.01; Figure 6a and Appendix A, https://doi.org/10.6084/m9.figshare.27134901).

FunCat analysis for the DEGs in the presence of CAS showed the transcriptional downregulation of genes coding for proteins involved in secondary metabolism and the upregulation of transporters and secondary metabolism (*p*-value < 0.01; Figure 6b and Appendix A, https://doi.org/10.6084/m9.figshare.27134901).

We evaluated the secondary metabolite production in these two clinical isolates by analyzing 4-day-cultured supernatants (grown in MM for 24 h and then supplemented with 2 µg/mL of CAS for 96 h at 37 °C) by LC-HRMS analysis (Figure 6c). Several SMs, such as demethoxyfumitremorgin C, trypostatin B, and spirotrypostatin A showed higher levels in the presence of CAS in the CM7555 than in the IFM61407 clinical isolate (Figure 6c). In contrast, other compounds such as brevianamide F, pseurotin, monomethyl sulochrin, and pyripyropene F were more strongly induced in the presence of CAS in IFM61407 than in the CM7555 clinical isolate (Figure 6c).

Taken together, these results suggest that the decreased CAS susceptibility of CM7555 clinical isolate is due to an increased response in the CWI pathway and PKA activity. There is some contribution made by the oxylipin pathway, but it does not seem to have the strongest influence on the difference in susceptibility to CAS. Once more, we observed a correlation between the modulation of the genes encoding proteins related to carbohydrate and SM metabolism and CAS susceptibility.

## 4. Discussion

There are very few antifungal drugs available against *A. fumigatus* infections. One of them, CAS, is a fungistatic drug used as second-line therapy. Our group has been investigating the mechanisms of CAS tolerance and how to potentiate its activity [23,34,53,54,55,56,57,58]. There is a close connection between CAS tolerance mechanisms and the activation of several signal transduction pathways, such as the CWI pathway, calcium/calmodulin/calcineurin, and protein kinase A [23,47,48,49]. We have previously shown that the treatment of *A. fumigatus* reference strains Af293 and CEA17 with its endogenous signaling oxylipin 5,8-diHODE resulted in the same three morphological responses being elicited by CAS (hyperbranching, increased septation, and increased chitin content in the cell wall) but without the occurrence of tip lysis [59]. Recently, we demonstrated that CAS tolerance in both reference strains of *A. fumigatus* is mediated through the inducible oxylipin signal 5,8-diHODE [24]. We have shown that the *A. fumigatus* response to 5,8-diHODE (produced by PpoA) requires TF ZfpA. It is an effector of CAS-mediated lysis since Δ*zfpA* is hypersusceptible to caspofungin, while the overexpression of ZfpA resulted in germlings that were highly tolerant to caspofungin [24,59]. Interestingly, the protective response to 5,8-diHODE does not require the TF CrzA, but the survival of the Δ*mpkA* mutant in the presence of CAS was still significantly improved by 5,8-diHODE [24]. In summary, we proposed that CAS and 5,8-diHODE synergistically activate chitin biosynthesis, aiming to construct a stronger cell wall with hyphal tips that are resistant to CAS-mediated lysis [24].

Here, we extended these studies by directly investigating the participation of *ppo* genes in CAS tolerance and concentrating our studies on the reference strain Af293. PpoA and PpoC have a heterogeneous and complex involvement in CAS tolerance and CPE. We focused our efforts on understanding the roles of PpoA and PpoC by using several different strategies. PpoA and PpoC are involved in the modulation of MpkA phosphorylation and PKA activity upon CAS exposure. This reinforces the idea of the control of the CWI pathway by PpoA and introduces the idea that oxylipins also modulate PKA activity. Actually, we have shown that PKA and the high-osmolarity glycerol (HOG) response pathways cooperatively control cell wall carbohydrate mobilization in *A. fumigatus* [47]. It is not clear if PpoA and PpoC are directly phosphorylated by either MpkA and/or PKA. The PpoA and PpoC phophosphorylation sites (as predicted by https://services.healthtech.dtu.dk/services/NetPhos-3.1/, access on 29 Sptember 2024 showed 4 and 16 sites for MpkA (p38MAPK) and 1 and 17 sites for PKA for PpoA and PpoC, respectively (Appendix A). It remains to be determined if these PpoA and PpoC residues are functional and are phosphorylated by MpkA and PKA.

Surprisingly, the transcriptional profiling of *ppoA* mutants using RNAseq did not provide a significant number of DEGs, but RNAseq analysis with *ppoC* mutants indicated category enrichment for those DEGs related to SM production and carbohydrate metabolism. Both MpkA and PKA are important for SM production and carbohydrate metabolism and the fact that PpoC is modulating these pathways could suggest that oxylipins are important for controlling these pathways. Considering that glucose-6-phosphate (G6P) is the precursor for UDP-N-acetylglucosamine (UDP-N-GlcNAc), the main building block of fungal chitin [60], it is possible that the effect on carbohydrate metabolism is impacting the activation of chitin biosynthesis. The control of SMs by oxylipins raises the possibility that oxylipins could also modulate other pathways that are related to fungal defense and/or signaling, a finding that complements previous studies on other Aspergilli [51,52].

We also investigated the differential expression of proteins in the Δ*ppoA* and Δ*ppoC* mutants exposed to CAS. Once more, category enrichment revealed the involvement of proteins related to secondary and carbohydrate metabolism. Additionally, we identified several proteins that are involved in calcium metabolism and in the CWI pathway. More interestingly, the regulation of the production of many of these proteins is concomitantly affected by both PpoA and PpoC, suggesting that both proteins are collaborating during pathway modulation. We validated the growth of the null mutants of several of the genes encoding TFs and signal transduction proteins in the presence of CAS and identified some mutants that have an altered susceptibility to CAS. It is currently not known how oxylipins induce the *A. fumigatus* downstream pathways. Oxylipins are produced intracellularly and later secreted. They could be reabsorbed in different parts of the hyphae and affect signaling through binding to endogenous or surface receptors. Interestingly, we identified two G-protein coupled receptors, GprC (AFUA_7G04800) and NopA (AFUA_7G01430), as being differentially expressed in the *ppo* mutants. Further characterization of these mutants is necessary to confirm if they are oxylipin effectors and if the GPCR proteins could be direct targets for oxylipins.

Aiming to characterize whether other clinical isolates that are different from our reference strains, Af293 and CEA17, also have their CAS susceptibility impacted by oxylipins, we characterized two clinical isolates, CM7555 (CAS-resistant) and IFM41607 (CAS-sensitive), which have extreme CAS sensitivity and CAS resistance. Surprisingly, IFM61407 is CAS-sensitive but still can produce oxylipins in the presence of CAS, while CM7555 is highly CAS-resistant and cannot produce oxylipins in the presence of CAS. As expected, the CWI pathway is more fully activated in the CM7555 strain since this strain is more resistant to cell-wall-damaging agents and shows increased MpkA phosphorylation and PKA activity. These results suggest that although oxylipins are important as modulators of CAS tolerance, they are not the only mechanism involved in this process. We need to assess the relationship between CAS tolerance and oxylipin production in a more significant number of clinical isolates to establish the contribution made by oxylipins to this process. We sequenced the genomes of both strains, aiming to identify possible differences that could help to explain additional mechanisms of CAS tolerance. We identified several statistically significant SNPs that are associated with CAS susceptibility and their further characterization will help to establish if these genes are involved in CAS susceptibility. We also exposed these two strains to CAS and investigated their transcriptional expression using RNAseq. Once more, carbohydrate metabolism and secondary metabolites are differentially expressed in both strains.

Antifungal drug resistance has been extensively studied [4]. However, drug tolerance and persistence are widely described for bacteria and are increasingly being examined for yeast-like organisms, but they are poorly defined for filamentous fungi [54]. Recently, we demonstrated that because all spores derived from an individual strain were phenotypically indistinct with respect to CPE, it is likely that CPE is a genetically encoded adaptive trait that should be considered an antifungal-tolerant phenotype [54]. Herein, we clearly demonstrated that CAS tolerance is a very complex and multifactorial process that involves not only oxylipins but also the modulation of several other signaling pathways. The study of which pathways are prevalent and how they are integrated could help us to understand how fungi can cope with stressing situations; this study can contribute to new strategies of antifungal drug development that could minimize the impact of antifungal drug resistance.

## Figures and Tables

**Figure 1 jof-10-00749-f001:**
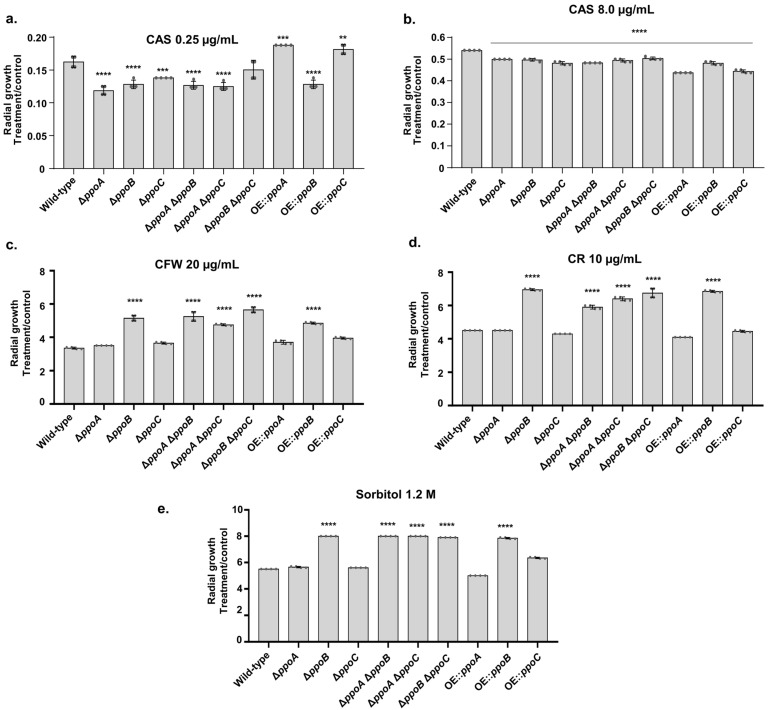
*A. fumigatus ppo* mutants exhibit altered susceptibility to CAS, cell-wall-damaging agents, and osmotic stress. The *A. fumigatus* wild-type and *ppo* mutants were grown for 5 days at 37 °C in MM that was supplemented or not with CAS 0.25 µg/mL (**a**), CAS 8 µg/mL (**b**), CFW 20 µg/mL (**c**), CR 10 µg/mL (**d**), and sorbitol 1.2 M (**e**). The results are expressed as the radial growth of the treatment, divided by the radial growth of the control, and are the average of three biological repetitions ± standard deviation. Statistical analysis was performed using Dunnett’s multiple comparisons test. ** *p*-value < 0.01, *** *p*-value < 0.001, **** *p*-value < 0.0001.

**Figure 2 jof-10-00749-f002:**
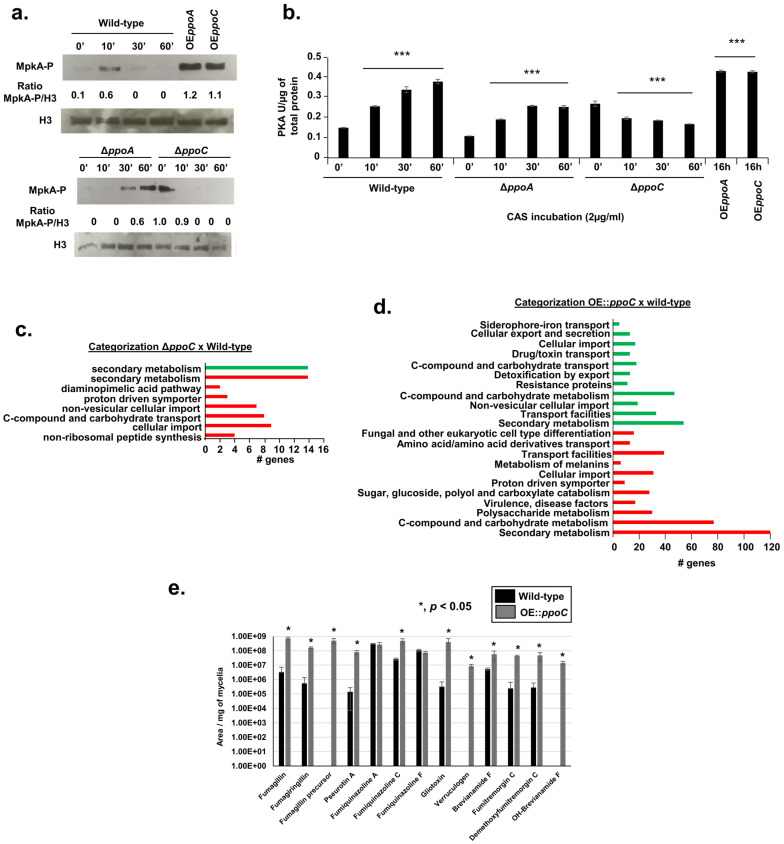
Cell wall integrity pathway and protein kinase A activity are important for *ppo*-altered susceptibility to CAS. (**a**) Western blot analysis for wild-type *A. fumigatus*, Δ*ppoA*, and Δ*ppoC*, grown in liquid MM for 16 h at 37 °C and transferred, or not, to 2 µg/mL of CAS for 0, 10, 30, and 60 min. OE*ppoA* and OE*ppoC* were grown in liquid MM for 16 h at 37 °C. MpkA-P was identified using P-p44/42 antibody, while histone H3 was identified by the anti-histone H3 antibody. (**b**) PKA activity of the wild-type and *ppo* mutants. The strains were grown as described in (**a**) and the samples were processed according to the PKA colorimetric activity kit (https://www.thermofisher.com/order/catalog/product/br/pt/EIAPKA, accessed on 29 September 2024) and normalized using the total protein. The results are the average of three repetitions ± standard deviation. *** *p*-value < 0.001 (**c**,**d**) FunCat analysis of the DEGs that were up- and downregulated in the Δ*ppoC* and OE*ppoC* mutants. (**e**) LC-HRMS analysis of the supernatant of the wild-type and OE*ppoC* strains grown in liquid MM for 5 days at 37 °C. The results are the average of three repetitions ± standard deviation.

**Figure 3 jof-10-00749-f003:**
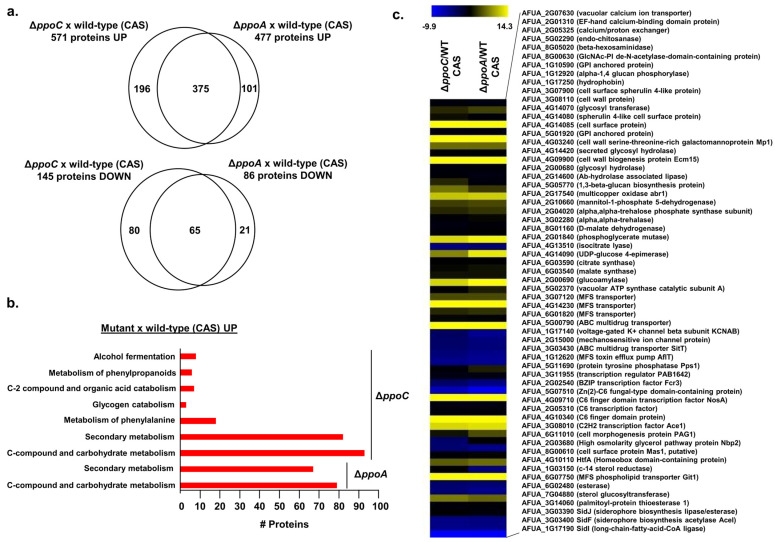
Proteomics for the wild type, Δ*ppoA*, and Δ*ppoC*. The wild type, Δ*ppoA*, and Δ*ppoC* strains were grown in liquid MM for 16 h at 37 °C and transferred to MM supplemented or not with 2 µg/mL of caspofungin for an additional 2 h. We used label-free quantitative proteomics to investigate those proteins that were differentially expressed. (**a**) Venn diagram showing the up- and downregulated proteins that are produced and shared or not by the Δ*ppoA* and Δ*ppoC* mutants. (**b**) FunCat categorization analysis, showing the differentially expressed proteins upregulated in the Δ*ppoA* and Δ*ppoC* mutants. (**c**) Heat map depicting the log2 fold change of differentially expressed proteins. Hierarchical clustering was performed in MeV (https://mev.tm4.org/, accessed on 29 September 2024) using Pearson’s correlation with complete linkage clustering. The results are the average of three independent biological repetitions.

**Figure 4 jof-10-00749-f004:**
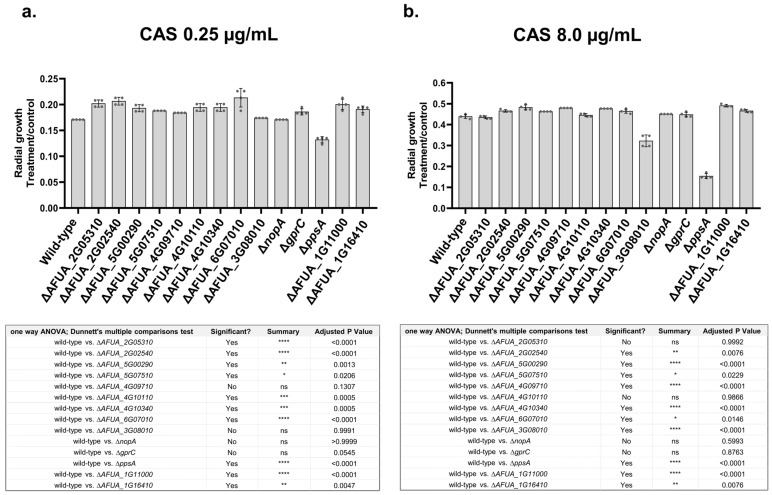
Growth phenotypes of *A. fumigatus* null mutants. (**a**) Inhibitory CAS concentrations of 0.25 µg/mL. (**b**) CPE concentrations of CAS of 8 µg/mL. The wild-type and the mutants were grown in MM or MM + CAS for 5 days at 37 °C. The results are expressed as radial growth on MM + CAS divided by MM, and are the average of three independent biological repetitions ± the standard deviation of three repetitions. Statistical analysis was performed using a one-way ANOVA with Dunnett’s multiple comparisons test. The corresponding *p*-values are shown in the embedded graph.

**Figure 5 jof-10-00749-f005:**
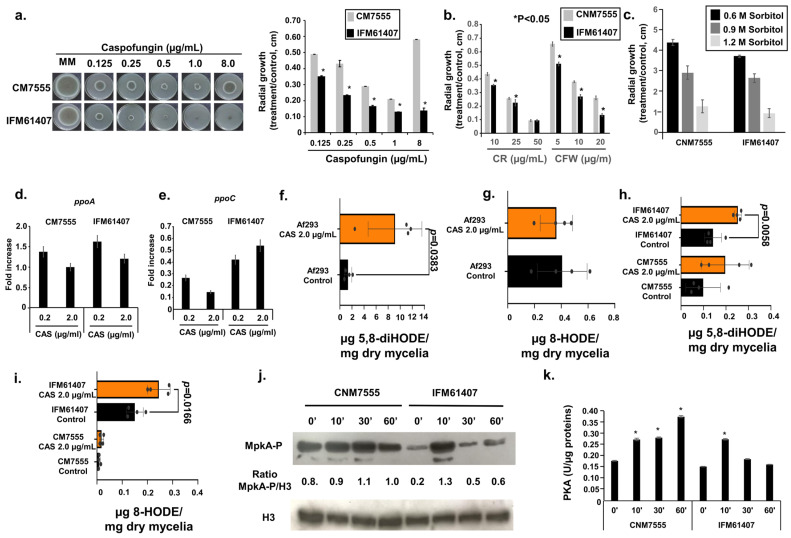
Characterization of the clinical isolates CM7555 and IFM61407. (**a**–**c**) Radial growth in the CM7555 and IFM61407 clinical isolates grown for 5 days at 37 °C on solid MM and MM + different concentrations of CAS, CR, CFW, or sorbitol. The results are expressed as the radial growth of the treatment, divided by the radial growth of the control, and are the average of three independent biological repetitions ± standard deviation. (**d**,**e**) RTqPCR analysis for the *ppoA* and *ppoC* genes. CM7555 and IFM61407 were grown for 16 h in liquid MM at 37 °C and transferred or not to MM + different CAS concentrations for 1 h. The results are the average of three independent biological repetitions ± standard deviation. (**f**–**i**) Oxylipin production, as measured by ultra-high-pressure liquid chromatography–high resolution mass spectrometry (UHPLC-HRMS). Af293, CM7555, and IFM61407 were grown for 24 h in liquid MM at 37 °C and transferred to MM + 2 µg/mL for an additional 48 h. (**j**) Western blot analysis for MpkA phosphorylation. The same experimental designs as for (**d**,**e**) were used for this experiment. (**k**) The strains were grown as described in (**j**), and the samples were processed according to the PKA colorimetric activity kit (https://www.thermofisher.com/order/catalog/product/br/pt/EIAPKA, accessed on 29 September 2024) and normalized using the total protein. The results are the average of three independent biological experiments ± standard deviation.

**Figure 6 jof-10-00749-f006:**
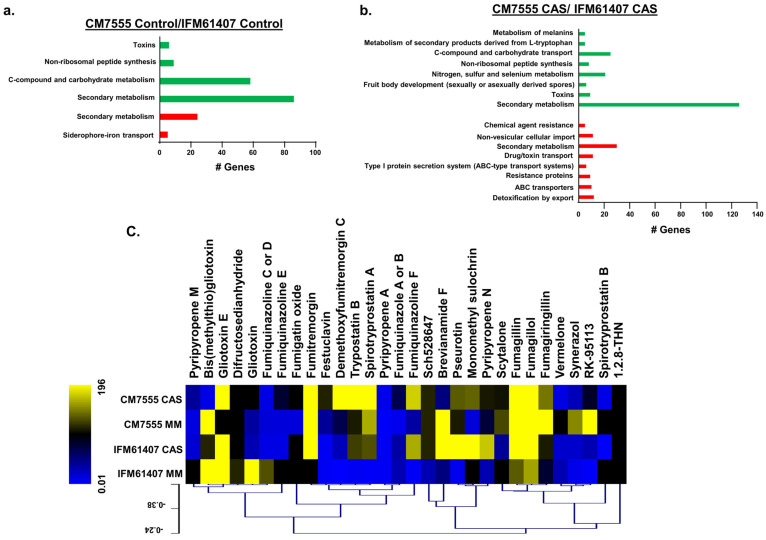
Differentially expressed genes in CM7555 and IFM61407 clinical isolates are enriched for secondary metabolite and carbohydrate metabolism. (**a**,**b**) FunCat categorization analysis, showing the differentially expressed genes up- and downregulated in CM7555/IFM61407 clinical isolates in the presence and absence of CAS. (**c**) Heat map depicting the relative quantification divided by dry weight of differentially expressed secondary metabolites identified by mass spectrometry. Hierarchical clustering was performed in MeV (htpp://mev.tm4.org/, accessed on 29 September 2024), using Pearson’s correlation with complete linkage clustering. The results are the average of three independent biological repetitions.

**Table 1 jof-10-00749-t001:** List of proteins specifically modulated in the ΔppoA or ΔppoC mutants when compared with the wild type in the presence of caspofungin (CSP).

Protein	Δ*ppoA*/WT (CSP) *
AFUA_5G09580 (hydrophobin)	5.7621
AFUA_1G13670 (related to sporozoite surface protein 2)	3.3037
AFUA_7G01430 (NopA, Opsin)	3.9007
AFUA_1G02800 (chitinase)	1.5652
AFUA_6G08470 (glycerol kinase)	1.982
AFUA_1G05580 (glycogenin)	1.572
AFUA_8G02060 (N-acetylglucosaminylphosphatidylinositol deacetylase)	1.0966
AFUA_6G09930 (AP-1-like transcription factor yap1)	1.0689
AFUA_2G05730 (MFS siderochrome iron transporter C)	1.0328
AFUA_6G07270 (long-chain fatty acid transporter)	1.0156
AFUA_2G02520 (Cell polarity protein Tea1)	−1.3091
AFUA_6G10480 (Peptidyl-prolyl cis-trans isomerase-like 3)	−1.9383
AFUA_4G06330 (C6 transcription factor)	−12.169
**Protein**	**Δ*ppoC*/WT (CSP) ***
AFUA_4G10130 (alpha-amylase)	12.271
AFUA_1G09540 (vacuolar transporter chaperon Vtc1)	11.797
AFUA_7G06750 (phosphoglycerate mutase)	9.2397
AFUA_7G04020 (lipase)	8.2124
AFUA_8G05360 (FAS1 domain-containing protein)	8.0016
AFUA_5G09130 (polysaccharide deacetylase)	7.1757
AFUA_1G12040 (chitin synthase export chaperone)	6.4859
AFUA_8G00230 (verruculogen synthase)	4.8539
AFUA_2G17580 (scytalone dehydratase arp1)	4.2638
AFUA_2G17530 (laccase abr2)	4.2588
AFUA_1G16410 (C6 transcription factor)	2.3637
AFUA_1G11000 (C6 transcription factor)	2.1857
AFUA_2G17560 (hydroxynaphthalene reductase arp2)	1.4489
AFUA_6G00750 (pyruvate decarboxylase)	1.4218
AFUA_5G12550 (N-acetylglucosaminylphosphatidylinositol deacetylase)	1.3624
AFUA_6G01840 (C6 transcription factor)	1.3588
AFUA_5G05600 (forkhead transcription factor Sep1)	1.3116
AFUA_1G17080 (glucosamine 6-phosphate N-acetyltransferase)	1.2597
AFUA_3G08820 (glycosyl hydrolase)	1.0264
AFUA_3G01530 (Phosphatidylglycerol specific phospholipase)	1.3353
AFUA_3G09950 (phospholipid:diacylglycerol acyltransferase)	1.2213
AFUA_6G07570 (palmitoyltransferase swf1)	1.213
AFUA_4G04200 (myo-inositol-1(Or 4)-monophosphatase)	1.1101
AFUA_1G09670 (HLH transcription factor GlcD gamma)	1.0999
AFUA_5G09070 (glycosyltransferase family 8 protein)	1.0634
AFUA_2G11860 (lipoprotein)	1.0475
AFUA_4G12180 (C6 transcription factor Prf)	−1.076
AFUA_2G04210 (L-tyrosine degradation gene cluster protein hmgX)	−1.2644
AFUA_3G00370 (phosphoketolase)	−1.4249
AFUA_1G09940 (casein kinase II beta 2 subunit)	−2.112
AFUA_7G08350 (alpha-1,3-glucanase)	−2.4887
AFUA_7G04800 (G-protein coupled receptor GprC)	−5.5981
AFUA_4G06940 (sphingolipid 4-desaturase)	−9.4216

* Differentially expressed proteins were defined as those with a minimum two-fold change in protein abundance (log_2_FC ≥ 1.0 and ≤−1.0; FDR of 0.05) when compared to the wild-type strain in the presence of CAS.

**Table 2 jof-10-00749-t002:** Main data about the genome sequencing of two *A. fumigatus* clinical isolates.

	Nuclear Genome					Mitogenome	
Assembly	Contigs	Largest Contig	Total Length	GC (%)	N50	Total Length	GC (%)
Af293	8	4918979	29384958	49.80	3948441	31765	25.38
CM7555	422	592903	28732111	49.53	210887	31750	25.38
IFM61407	998	982941	30706367	48.74	228020	31751	25.38

**Table 3 jof-10-00749-t003:** Characterization of the SNPs associated with CAS susceptibility.

CM7555	IFM61407	Chromosome	Position	Ref	Alt	Gene	Variant Type	Function
	X	NC_007195.1	1182496	G	A	Afu2g04280	upstream_gene_variant	asparaginase, putative
X		NC_007196.1	3242018	T	C	Afu3g12290	5_prime_UTR_variant	pre-mRNA splicing factor Dim1
X		NC_007196.1	3242454	A	C	Afu3g12300	5_prime_UTR_variant	60S ribosomal protein L22, putative
	X	NC_007196.1	3511190	C	T	Afu3g13230	missense_variant	AT DNA binding protein, putative
X		NC_007196.1	3512039	G	A	Afu3g13230	missense_variant	AT DNA binding protein, putative
	X	NC_007196.1	3514244	T	C	Afu3g13230	upstream_gene_variant	AT DNA binding protein, putative
	X	NC_007196.1	3519307	G	A	Afu3g13260	synonymous_variant	Rad2-like endonuclease, putative
	X	NC_007196.1	3520699	G	T	Afu3g13260	synonymous_variant	Rad2-like endonuclease, putative
	X	NC_007196.1	3528767	C	T	Afu3g13300	missense_variant	hypothetical protein
	X	NC_007196.1	3543581	T	A	Afu3g13390	3_prime_UTR_variant	vacuolar ATP synthase subunit d, putative
	X	NC_007196.1	3546410	C	T	Afu3g13400	synonymous_variant	nucleolar protein nop5
	X	NC_007198.1	2400034	T	C	Afu5g09320	synonymous_variant	signal transduction protein Syg1, putative
	X	NC_007198.1	2799527	G	A	Afu5g10940	upstream_gene_variant	hypothetical protein
X	X	NC_007198.1	3545720	A	G	Afu5g13510	missense_variant	cell cycle control protein (Cwf8), putative
X		NC_007199.1	1516436	C	T	Afu6g06880	3_prime_UTR_variant	negative regulator of DNA transposition (Rtt106), putative
	X	NC_007199.1	1736813	T	C	Afu6g07640	synonymous_variant	lysyl-tRNA synthetase
X		NC_007200.1	371671	C	T	Afu7g01430	upstream_gene_variant	opsin, putative
X	X	NC_007200.1	374641	T	C	Afu7g01440	missense_variant	hypothetical protein
X		NC_007200.1	409263	C	A	Afu7g01560	synonymous_variant	FAD dependent oxidoreductase, putative
X	X	NC_007200.1	532236	T	A	Afu7g01970	missense_variant	RTA1 domain protein, putative

## Data Availability

The raw reads generated by Illumina were deposited on the NCBI’s GenBank database under the following SRA IDs: SRR28432704 to SRR28432733 (BioProject accession PRJNA1091189). The mass spectrometry proteomics data have been deposited at the ProteomeXchange Consortium via the PRIDE [61] partner repository, with the dataset identifier PXD054599 (Project Name: The influence of Aspergillus fumigatus fatty acid oxygenases PpoA and PpoC on caspofungin susceptibility; Project accession: PXD054599). Reviewer account details: Username: reviewer_pxd054599@ebi.ac.uk; Password: AhWy4h01YNXB.

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
