# Peer review of "The Influence of Aspergillus fumigatus Fatty Acid Oxygenases PpoA and PpoC on Caspofungin Susceptibility"

_jof, 2024, doi:10.3390/jof10110749_

Round 1
Reviewer 1 Report
Why does the ΔppoB ΔppoC double mutant not affect caspofungin susceptibility, while single deletions of either b or c increase susceptibility? Please explain.
Have you tested the effect of single deletions of a, b, or c on the transcription levels of the other two genes?
Lines 412-414: Does the MPKA PKA signaling pathway regulate PpoA and PpoC directly or indirectly? Do PpoA and PpoC proteins have phosphorylation sites?
Why doesn't Figure 1 include a spot assay? Adding it would make the results more intuitive and credible, enhancing persuasiveness.
Author Response
Major comments
Why does the ΔppoB ΔppoC double mutant not affect caspofungin susceptibility, while single deletions of either b or c increase susceptibility? Please explain.
Answer: We thank the reviewer for the comment. We do not know why this is happening, but we added some speculation to the text (lines 386 to 389): “Surprisingly, the double mutant ΔppoB ΔppoC is as sensitive to CAS as the wild-type while ΔppoB and ΔppoC are more sensitive than the double mutant (Figure 1a). It is possible there are additional mechanisms of coping with CAS stress that are activated when both genes are deleted and this remains to be determined.”
Have you tested the effect of single deletions of a, b, or c on the transcription levels of the other two genes?
Answer: We thank the reviewer for the question. We have not checked it. However, we investigated the ppoA and ppoB mRNA accumulation in the RNAseq data and this was added to the manuscript (lines 462 to 465): ”We have not observed ppoA, ppoB, and ppoC differential expression in both ΔppoC and OEppoC DEGs (Supplementary Tables 1 and 2), suggesting PpoC altered levels are not affecting the differential expression of these two genes”.
Lines 412-414: Does the MPKA PKA signaling pathway regulate PpoA and PpoC directly or indirectly? Do PpoA and PpoC proteins have phosphorylation sites?
Answer: We thank the reviewer for the comment. We have added an analysis of the predicted phosphorylation sites for these two kinases in the PpoA and PpoC (lines 687 to 693): “It not clear if PpoA and PpoC are directly phosphorylated by either MpkA and/or PKA. PpoA and PpoC phophosphorylation sites (as predicted by https://services.healthtech.dtu.dk/services/NetPhos-3.1/) showed 4 and 16 sites for MpkA (p38MAPK) and 1 and 17 sites for PKA for PpoA and PpoC, respectively (Supplementary Text 1). It remains to be determined if these PpoA and PpoC residues are functional and are phosphorylated by MpkA and PKA.”
Detailed comments
Why doesn't Figure 1 include a spot assay? Adding it would make the results more intuitive and credible, enhancing persuasiveness.
Answer: We thank the reviewer for the suggestion. We used radial growth because this methodology allows the application of statistics. If the reviewer does not mind, we would prefer to keep the data as it is now, i.e., radial growth analysis.
Reviewer 2 Report
The manuscript by Delbaje et al. focuses on the influence of Aspergillus fumigatus fatty acid oxygenases PpoA and PpoC on Caspofungin susceptibility. The experiments are well-designed, and the manuscript is well-written. However, the objectives of the manuscript are not well defined, and some results are not well explained/discussed.
Major points:
1. In fig1, the authors show the CAS susceptibility and CPE for all single and double mutants. However, later on, they mentioned that DppoB and OEppoB are not being further investigated. Why? What is the relationship between the expression of ppoB to ppoA and ppoC? If they have an antagonistic relationship, studying ppoB should strengthen the conclusion. The exclusion is not justified.
2. Figure 2, the figure legend says that “actin by anti-histone H3 antibody”. Just to clarify, the authors are using antibody against histone to measure the levels of actin. Are H3 levels in the figures actin levels?
3. The protein ratio is presented as MpkA-P/H3. If they used antibody against total MpkA and MpkA-P, where are the results for MpkA Ab? Why is the ratio not presented for P/total MpkA protein?
4. Line 647-650, the authors infer that MpkA and PKA are important for SM production and the fact that PpoC is modulating these pathways could suggest oxylipins are important for controlling these pathways. This is a stretch. The authors have themselves concluded that there are multiple pathways contributing to these responses, as evident from clinical isolates, authors have not presented any data to back up that claim. If this is a hypothesis, the authors should explain the other side of the results as well.
5. The authors say that studying how fungi cope with stressing situations can contribute to new strategies of anti-fungal development. What class? Static or cidal? Improve on caspofungin or echinocandin class? This is a stretch. How studying CPE for once class affects other class. The authors themselves mentioned three classes exist which are very different from one another.
6. Authors mentioned that to create the OE strains, they amplified ~1kb upstream and downstream of the protein coding sequence and this was fused with pyrG::gpdA from pJMP9.1. What is the reference for the source of the plasmid? Does the fusion happen at the transcription start site of the gpdA promoter or the translational start site?
7. How many cycles were completed for semi-quantitative PCR? Is there a image analysis to show the levels of the genes and calculate their OE?
Minor comments:
1. Line 51, change the word colossal. When a more specific number is provided in an example why use a non-specific term?
2. Line 64-65, Healthy individuals are becoming susceptible after contracting COVID-19. Someone who has contracted Covid 19 will be considered healthy? We know virus suppresses the immune system and should fall in the category of immune suppressive categories
3. References for pJMP and Af293 host strains should be provided
Author Response
The manuscript by Delbaje et al. focuses on the influence of Aspergillus fumigatus fatty acid oxygenases PpoA and PpoC on Caspofungin susceptibility. The experiments are well-designed, and the manuscript is well-written. However, the objectives of the manuscript are not well defined, and some results are not well explained/discussed.
Answer: We thank the reviewer for the suggestions and comments.
Major points:
- In fig1, the authors show the CAS susceptibility and CPE for all single and double mutants. However, later on, they mentioned that DppoB and OEppoB are not being further investigated. Why? What is the relationship between the expression of ppoB to ppoA and ppoC? If they have an antagonistic relationship, studying ppoB should strengthen the conclusion. The exclusion is not justified.
Answer: We thank the reviewer for the question. The function and enzymatic activity of PpoB are not well-understood and that was the main reason why we decided to investigate PpoA and PpoB. There are also previous publications of our group showing the importance of PpoA and PpoC in hyperbranching and chitin metabolism. Although PpoB was not investigated here, we considered to work on it in a near future. We have not checked the expression of these three genes at the different ppo mutants background. However, ppoA, ppoB, and ppoC mRNA accumulation is not different in the wild-type, ΔppoC and OEppoC (lines 462 to 465): “We have not observed ppoA, ppoB, and ppoC differential expression in both ΔppoC and OEppoC DEGs (Supplementary Tables 1 and 2), suggesting PpoC altered levels are not affecting the differential expression of these two genes”.
- Figure 2, the figure legend says that “actin by anti-histone H3 antibody”. Just to clarify, the authors are using antibody against histone to measure the levels of actin. Are H3 levels in the figures actin levels?
Answer: We thank the reviewer for noticing it and we apologize about the mistake. Actually, this is “histone H3 by anti-histone H3 antibody” and this was now changed in the legend of the Figure 2.
- The protein ratio is presented as MpkA-P/H3. If they used antibody against total MpkA and MpkA-P, where are the results for MpkA Ab? Why is the ratio not presented for P/total MpkA protein?
Answer: We thank the reviewer for the question. Unfortunately, in our hands the commercial antibody against total MpkA does not work well, recognizing several no specific bands. That is the reason why we are normalizing the experiment dividing the MpkA-P by histone H3.
- Line 647-650, the authors infer that MpkA and PKA are important for SM production and the fact that PpoC is modulating these pathways could suggest oxylipins are important for controlling these pathways. This is a stretch. The authors have themselves concluded that there are multiple pathways contributing to these responses, as evident from clinical isolates, authors have not presented any data to back up that claim. If this is a hypothesis, the authors should explain the other side of the results as well.
Answer: Unfortunately, we disagree about this conclusion raised by the reviewer. The fact that secondary metabolites are modulated by oxylipin is a strong argument about the impact of oxylipins on MpkA and PKA activities. Many (if not most) of the secondary metabolite production is dependent on these two pathways.
- The authors say that studying how fungi cope with stressing situations can contribute to new strategies of anti-fungal development. What class? Static or cidal? Improve on caspofungin or echinocandin class? This is a stretch. How studying CPE for once class affects other class. The authors themselves mentioned three classes exist which are very different from one another.
Answer: Once more, unfortunately we disagree with the reviewer. Our group and other groups around the world have demonstrated that drug repurposing can help to synergize both echinocandins and azoles by affecting primarily the cell wall integrity pathway and cell membrane permeability. In the case of echinocandins, in the presence of synergizers, these drugs can be converted into fungicidal. In the case of fungicidal drugs, like azoles, the synergizers can bypass azole-resistance. The definition of stressing pathways that are activated by echinocandins and azoles can help to identify new adjuvants that synergize these drugs. Please, see:
- a) Dos Reis TF, Diehl C, Pinzan CF, de Castro PA, Goldman GH. Brilacidin, a host defense peptide mimetic, potentiates ibrexafungerp antifungal activity against the human pathogenic fungus Aspergillus fumigatus. Microbiol Spectr. 2024 Aug 6;12(8):e0088824. doi: 10.1128/spectrum.00888-24. Epub 2024 Jul 9. PMID: 38980033; PMCID: PMC11302226.
- b) Dos Reis TF, de Castro PA, Bastos RW, Pinzan CF, Souza PFN, Ackloo S, Hossain MA, Drewry DH, Alkhazraji S, Ibrahim AS, Jo H, Lightfoot JD, Adams EM, Fuller KK, deGrado WF, Goldman GH. A host defense peptide mimetic, brilacidin, potentiates caspofungin antifungal activity against human pathogenic fungi. Nat Commun. 2023 Apr 12;14(1):2052. doi: 10.1038/s41467-023-37573-y. PMID: 37045836; PMCID: PMC10090755.
- Authors mentioned that to create the OE strains, they amplified ~1kb upstream and downstream of the protein coding sequence and this was fused with pyrG::gpdA from pJMP9.1. What is the reference for the source of the plasmid? Does the fusion happen at the transcription start site of the gpdA promoter or the translational start site?
Answer: The reference source for the plasmid was added to the reference list. The fusion is at translational start site and this information was added to the text (line 158): “All the overexpression fusions happened at translational start site”.
- How many cycles were completed for semi-quantitative PCR? Is there a image analysis to show the levels of the genes and calculate their OE?
Answer: We thank the reviewer for the question. We have added this data to the text and a supplementary figure (Lines 444 to 446): “The OEppoA and OEppoC have about 12- and 8-fold more ppoA mRNA accumulation and 17- and 6-fold (at 16 and 24 h), respectively, than the wild-type (Supplementary Figure 2).”
Minor comments:
- Line 51, change the word colossal. When a more specific number is provided in an example why use a non-specific term?
Answer: We thank the reviewer for the comment. The word was changed (line 53): “…produce great numbers of clonal spores…”
- Line 64-65, Healthy individuals are becoming susceptible after contracting COVID-19. Someone who has contracted Covid 19 will be considered healthy? We know virus suppresses the immune system and should fall in the category of immune suppressive categories
Answer: We thank the reviewer for the comment. The sentence was changed to (lines 67 to 68): “Therefore, individuals are becoming susceptible to this deadly Aspergillus infection once contracted COVID-19, which is now widely prevalent.”
- References for pJMP and Af293 host strains should be provided
Answer: We thank the reviewer for noticing it. The references were added to these plasmids and strains.
Reviewer 3 Report
This study investigates the role of fatty acid oxygenases in caspofungin (CAS) susceptibility in Aspergillus fumigatus. Moreover, it provides new insights into mechanisms of antifungal drug susceptibility and uses a combination of genetic, biochemical, and omics approaches to elucidate the multifaceted nature of CAS tolerance mechanisms in A. fumigatus.
1. Does the author have more evidence, such as protein identity or sequence alignment data, to support excluding ppoB from the CAS-susceptibility independent pathway, beyond just phenotypic information?
2. In Figure 2, some data showed major differences between the ppoA mutant and ppoC mutant strains (e.g., protein kinase expression or PKA activity). Can the authors identify whether ppoA or ppoC plays a more critical role in the proposed mechanism for CAS sensitivity?
3. Line 412: What is the "ppos"?
4. Regarding Figures 5D and 5E, the lack of statistically significant differences in the data is indeed an important point that should be addressed in the main results. If the differences in ppoA and ppoC between CM7555 and IFM61407 in response to various CAS concentrations are not statistically significant, this is a critical point to discuss. This result could be seen as inconclusive and may make it challenging to interpret the proposed pathway.
5. While two clinical isolates with contrasting CAS susceptibility were examined, this limited sample may not be representative of the diversity found in clinical A. fumigatus strains. This point warrants further discussion. Analyzing a larger panel of clinical isolates could provide more robust conclusions and better reflect the variability in CAS susceptibility among A. fumigatus strains encountered in clinical settings.
Author Response
Major comments
This study investigates the role of fatty acid oxygenases in caspofungin (CAS) susceptibility in Aspergillus fumigatus. Moreover, it provides new insights into mechanisms of antifungal drug susceptibility and uses a combination of genetic, biochemical, and omics approaches to elucidate the multifaceted nature of CAS tolerance mechanisms in A. fumigatus.
Answer: We thank the reviewer for the comments and suggestions.
Detailed comments
- Does the author have more evidence, such as protein identity or sequence alignment data, to support excluding ppoB from the CAS-susceptibility independent pathway, beyond just phenotypic information?
Answer: We thank the reviewer for the question. The function and enzymatic activity of PpoB is not well-understood and that was the main reason why we decided to investigate PpoA and PpoC. There are also previous publications of our group showing the importance of PpoA and PpoC in hyperbranching and chitin metabolism. Although PpoB was not investigated here, we considered to work on it in a near future.
- In Figure 2, some data showed major differences between the ppoA mutant and ppoC mutant strains (e.g., protein kinase expression or PKA activity). Can the authors identify whether ppoA or ppoC plays a more critical role in the proposed mechanism for CAS sensitivity?
Answer: We thank the reviewer for the comment. The function of ppoC is not well established. It is possible there is complementary or antagonic regulation of these pathways. Unfortunately, with the current results and analysis it is not possible to distinguish the individual contributions of PpoA and PpoC to these processes, although it seems PpoA plays a more active relationship in the activation of hyperbranching and septation.
- Line 412: What is the "ppos"?
Answer: We apologize about it. The sentence was changed to (line 434): “These results suggest that the ppo mutants are important for the response to CAS.”
- Regarding Figures 5D and 5E, the lack of statistically significant differences in the data is indeed an important point that should be addressed in the main results. If the differences in ppoA and ppoC between CM7555 and IFM61407 in response to various CAS concentrations are not statistically significant, this is a critical point to discuss. This result could be seen as inconclusive and may make it challenging to interpret the proposed pathway.
Answer: We thank the reviewer for the comment. Our results showed exactly that PpoA and PpoC are important but they are not the single mechanism of caspofungin tolerance in the clinical isolates. The results of the RTqPCR corroborate this finding that is already discussed in the Discussion section. As discussed below (an in the manuscript, lines 730 to 732), “We need to assess the relationship between CAS tolerance and oxylipin production in a more significant number of clinical isolates to establish the contribution of oxylipins in this process”.
- While two clinical isolates with contrasting CAS susceptibility were examined, this limited sample may not be representative of the diversity found in clinical A. fumigatus strains. This point warrants further discussion. Analyzing a larger panel of clinical isolates could provide more robust conclusions and better reflect the variability in CAS susceptibility among A. fumigatusstrains encountered in clinical settings.
Answer: We thank the reviewer for this comment. However, we have already added this comment in the manuscript (lines 730 to 732): “We need to assess the relationship between CAS tolerance and oxylipin production in a more significant number of clinical isolates to establish the contribution of oxylipins in this process”.